# Testing a first online intervention to reduce conformity to cyber aggression in messaging apps

**Daniëlle N. M. Bleize**[1]\*, **Doeschka J. Anschütz**[1], **Martin Tanis**[2], **Moniek Buijzen**[3]

**1** Behavioural Science Institute, Radboud University, Nijmegen, The Netherlands, **2** Department of Communication Science, Vrije Universiteit Amsterdam, Amsterdam, The Netherlands, **3** Erasmus School of Social and Behavioural Sciences, Erasmus University Rotterdam, Rotterdam, The Netherlands

\* danielle.bleize@ru.nl

**Data Availability Statement:** The data that support the findings of this study are openly available on the Open Science Framework (OSF) at https://bit.ly/2SYR1V1.

## Abstract

Early adolescents frequently use mobile messaging apps to communicate with peers. The popularity of such messaging apps has a critical drawback because it increases conformity to cyber aggression. Cyber aggression includes aggressive peer behaviors such as nasty comments, nonconsensual image sharing, and social exclusion, to which adolescents subsequently conform. Recent empirical research points to peer group norms and reduced accountability as two essential determinants of conformity to cyber aggression. Therefore, the current study aimed to counteract these two determinants in a 2 (peer group norms counteracted: yes, no) x 2 (reduced accountability counteracted: yes, no) design. We created four intervention conditions that addressed adolescents' deficits in information, motivation, and behavioral skills. Depending on the condition (peer group norms, reduced accountability, combination, or control), we first informed participants about the influence of the relevant determinant (e.g., peer group norms). Subsequently, participants performed a self-persuasion task and formulated implementation-intentions to increase their motivation and behavioral skills not to conform to cyber aggression. Effectiveness was tested with a messaging app paradigm and self-report among a sample of 377 adolescents ($M_{age}$ = 12.99, $SD_{age}$ = 0.84; 53.6% boys). Factorial ANCOVAs revealed that none of the intervention conditions reduced conformity to cyber aggression. Moreover, individual differences in susceptibility to peer pressure or inhibitory control among adolescents did not moderate the expected relations. Therefore, there is no evidence that our intervention effectively reduces conformity to cyber aggression. The findings from this first intervention effort point to the complex relationship between theory and practice. Our findings warrant future research to develop potential intervention tools that could effectively reduce conformity to cyber aggression.

## Introduction

Messaging apps such as WhatsApp have become tremendously popular among early adolescents worldwide [1–3]. Unfortunately, this has also led to increasing group-based cyber

**Funding:** The author(s) received no specific funding for this work.

**Competing interests:** The authors have declared that no competing interests exist.

aggression on these messaging apps. Cyber aggression includes all derogatory, offensive, or harmful online peer-to-peer behaviors [2–5]. It is, thus, a more comprehensive concept than cyberbullying–which researchers conceptualize more narrowly as repetitive, with intent to harm, and stemming from power imbalance [6–8]. Particularly problematic in messaging apps is that adolescents conform to each other's negative peer behaviors. An example of this is when members of a WhatsApp group share nasty comments about a peer among each other. Messaging apps are pre-eminently suited for such conformity behaviors, because these apps provide private communication channels with intimate friends or other social ties [9, 10]. However, being involved in cyber aggression poses significant risks for adolescents' well-being, such as poor academic performance [11], mental health problems [12–14], and even self-harm [6, 12, 15]. These risks call for intervention efforts to reduce conformity to cyber aggression in adolescents' messaging apps.

Typically adolescents agree that it is wrong to engage in cyber aggressive behaviors [16]. However, it can be difficult not to conform when these behaviors occur in adolescents' messaging app groups. It can be difficult to not conform because group conformity is a potent mechanism through which adolescents influence each other [17, 18]. Adolescents have a strong urge to belong to peer groups [5, 19]. Group conformity is associated with social rewards (e.g., social recognition from peers). Obtaining such social rewards typically increases adolescents' self-esteem [20]. Moreover, adolescents perceive those who engage in cyber aggression as more popular. Therefore, aggressive behaviors become a way to gain status among peers [18, 21]. For example, empirical work has shown that adolescents who bully are perceived as more popular by their peers over time [22, 23]. Furthermore, from theory in the field of developmental psychology we know that adolescents' immature cognitive development causes them to experience difficulties inhibiting (initial) behavioral responses [24, 25]. Receiving a private picture of a peer can bring feelings of excitement or sensation [26, 27], which can subsequently impede adolescents to reflect critically on the consequences of their behavior. Altogether, it may be difficult for adolescents not to conform to cyber aggressive behaviors when these occur in messaging apps. There is, thus, an urgent need for intervention efforts that help adolescents to not conform to cyber aggression [28].

The objective of the current study is to develop and assess the effectiveness of a behavior change intervention to reduce conformity to cyber aggression in early adolescents' messaging apps. Intervention programs so far have not specifically targeted the theoretical mechanisms related to conformity to harmful peer-to-peer behaviors. Additionally, only few focused on the unique context of messaging apps [3, 29]. Therefore, the development of our intervention consists of two steps. In the first step, we identify the critical determinants of conformity to cyber aggression in messaging apps. To do this, we will review recent empirical research on messaging app groups. In the second step, we systematically translate the relevant theoretical insights to an applied intervention. To develop this intervention, we will use a behavior change framework that has proven successful in eliciting positive behavior change: the Information-Motivation-Behavioral Skills framework [IMB framework, 30, 31].

## Theoretical background

### Empirical research on conformity to cyber aggression

The first step in developing our intervention is to review the empirical research on aggressive behavior in general, and on conformity to cyber aggression specifically. Over the past few decades, research on adolescent development has consistently revealed that peer influence plays an essential role in the occurrence of aggressive behaviors [18, 32, 33]. Recent literature and meta reviews also corroborate the role of peer influence [34, 35]. Adolescents are attuned

to positive regard from peers [18, 36] and look at their peers to understand which behaviors are acceptable and desired. Researchers refer to this susceptibility to peers as peer socialization [18]. Generally speaking, adolescents tend to conform to peer norms [5, 18], especially when it concerns close peers like friends [5, 32, 37, 38]. Key empirical studies have shown that conforming to the standards of a valued group allows adolescents to gain status in the group and helps them to obtain a positive sense of self [18, 20, 21].

Empirical studies on conformity to peer aggression have pointed to the relevance of two specific determinants of conformity: peer group norms and reduced accountability. First, studies show that aggression among adolescents increases when aggression is considered acceptable by peers [32, 33]. If peers are positive towards aggressive behaviors, other adolescents are more likely to engage in these behaviors as well. Second, studies show that aggressive peer behaviors occur more frequently when perpetrators are not held accountable for their behaviors (when they think their actions remain confidential from people outside of their peer group) [39–41]. A recent systematic review on the determinants and consequences of bystander interventions in cyberbullying found that perceived accountability boosts the likelihood that people will refrain from conforming to cyber aggression and instead help victims [42]. Adolescents who feel unaccountable to others typically perceive little risk of potential adult intervention or punishment for their behaviors [2, 43]. Consequently, adolescents are more likely to behave consistent with their peers' attitudes and behaviors.

Bleize et al. [44] examined whether peer group norms and reduced accountability affect conformity to cyber aggression in early adolescents' messaging apps. In two experiments, they found evidence supporting the relevance of these two determinants. Both experiments included a scripted WhatsApp conversation to test conformity to cyber aggressive group norms. The authors led participants to believe that they would have a conversation about cyber aggressive behaviors with some of their peers and that their reactions in the conversation would remain private or be made public. In reality, the authors preprogrammed the responses in the conversation. Participants replied to the preprogrammed responses of several ostensible peers. The results of the experiments showed that adolescents consistently conformed to cyber aggressive group norms. When peers indicated approval for cyber aggression (e.g., forwarding nasty messages about a peer), adolescents also became more positive towards the behavior [44]. Moreover, Bleize et al. [44] found support for the process of reduced accountability. When the researchers told adolescents that their responses in the messaging app group would remain private to others outside of the group, adolescents conformed more to cyber aggressive group norms (compared to when the authors told adolescents that they had to share their responses publicly in-class) [44]. In sum, Bleize et al.'s [44] findings confirm that peer group norms and reduced accountability influence conformity to cyber aggression.

## Applying a theoretical framework to intervention development

The second step in developing an intervention is to systematically translate the above-mentioned theoretical insights to an applied intervention. We used the Information-Motivation-Behavioral Skills framework [IMB framework, 30, 31] for this purpose. The IMB framework is a theoretical framework to help develop behavioral interventions. It was explicitly designed to be easy to translate to empirically targeted interventions and has proven successful in eliciting positive behavior change [e.g., tackling cyber aggression and cyberbullying, 45].

The IMB framework proposes that three key elements should be included in interventions to stimulate positive behavior change: information, motivation, and behavioral skills [30, 31]. First, adolescents must be *informed* of and understand the desired behavior. This means that adolescents should know what the behavior entails, understand its underlying processes, and

be aware of its consequences [31, 45]. According to the IMB framework, adolescents can only adapt their cyber aggressive behaviors if they can recognize them. Therefore, a successful intervention should inform adolescents about the determinants of cyber aggression (e.g., peer group norms and reduced accountability).

Second, adolescents must feel *motivated* to perform the target behavior. Due to the affective nature of and social dynamics involved in aggressive behavior, adolescents may not always feel motivated to refrain from cyber aggression. Adolescents might recognize that nonconsensual image-sharing is wrong [16], but may still want to share these images with their friends [27]. In the context of cyber aggression, adolescents may feel more motivated to engage in the desired behavior if it is for personally relevant reasons. Therefore, a successful intervention should motivate adolescents not to conform to cyber aggression, for example by using self-persuasion [46, 47].

Third, adolescents must have the right *behavioral skills* to resist conformity to cyber aggression [31, 45]. Even when adolescents are knowledgeable about cyber aggression and feel motivated to act on their knowledge, they still need the skills to act appropriately in such situations. Implementation-intentions could be used to overcome potential inertia in situations involving cyber aggression [48–50]. These if-then plans can help adolescents to better regulate their behavior. In the methods section of this manuscript, we will elaborate on how exactly the three key elements of the IMB model are implemented in our intervention.

## A behavior change intervention to reduce conformity in messaging apps

The objective of our study is to develop and assess the effectiveness of a behavior change intervention to reduce conformity to cyber aggression in early adolescents' messaging apps. On a theoretical level, we aim to develop a theory-based intervention grounded in recent insights on conformity to cyber aggression in messaging app groups. In addition, we aim to systematically translate these insights into an effective applied intervention. On a practical level, we aim to test whether our intervention is indeed a viable approach to reduce conformity to aggressive peer behaviors in messaging apps. To reach our objective, the intervention will consist of four conditions (peer group norms, reduced accountability, combination, and control). All conditions except the control include the three key IMB model components of information, motivation and behavioral skills, which are considered necessary for any successful intervention [30, 31]. The implementation of the information component differs in each condition, depending on the target determinant (peer group norms, reduced accountability, or both). The implementation of the motivation and behavioral skills components (i.e., self-persuasion and implementation-intentions) is similar in each condition.

We expect that the conditions focusing on either single process (peer group norms or reduced accountability) reduce conformity to cyber aggression compared to the control condition. Additionally, we expect a cumulative effect when these conditions are combined in the combination condition, because the peer group norms and reduced accountability conditions are both expected to reduce conformity. In other words, we expect the combination condition to be most effective. We examine the intervention conditions' effectiveness both directly after the intervention (Time 1) and four weeks post-intervention (Time 2). Specifically, our expectations are as follows:

H1: The condition targeting peer group norms reduces conformity to cyber aggression compared to the conditions not targeting peer group norms, both at Time 1 and Time 2.

H2: The condition targeting reduced accountability reduces conformity to cyber aggression compared to the conditions not targeting reduced accountability, both at Time 1 and Time 2.

H3: The combination condition (peer group norms + reduced accountability) is most effective at reducing conformity to cyber aggression, compared to the conditions focusing on either single process and the control condition, both at Time 1 and Time 2.

### The moderating role of adolescents' susceptibility to peer pressure and inhibitory control

In examining the effectiveness of the current intervention, it is also important to consider individual differences in adolescents' social-cognitive development. Differences in adolescents' social-cognitive development may magnify or mitigate potential intervention effects. Specifically, differences in susceptibility to peer pressure and inhibitory control could play a role. Although adolescents are typically most responsive to peer influence during early adolescence (ages 12–15), some are more susceptible than others [18, 21]. Susceptibility to peer pressure could moderate the intervention effects in two ways. On the one hand, the intervention may work better for adolescents with high susceptibility to peer pressure. Adolescents with high susceptibility to peer pressure typically have difficulties standing up to peers [18, 51], and, thus, might benefit most from an intervention that helps them to stand up to their peers. Contrarily, the intervention may work better for adolescents with low susceptibility to peer pressure. Adolescents with low susceptibility to peer pressure are usually able to resist peer influence [18], but might not yet have learned the right behavioral skills to resist conformity to cyber aggression.

Second, we consider how individual differences in inhibitory control may affect the intervention's effectiveness. Adolescents differ in their ability to inhibit behavioral responses [18]. When inhibitory control is low, adolescents typically have difficulties controlling their reactions to emotional situations [52] such as those involving cyber aggression. Again, this may moderate the intervention effects in two ways. The intervention might work better for adolescents with low inhibitory control, because they may benefit most from an intervention helping them better regulate their behaviors [53]. Then again, the intervention could work better for adolescents with high inhibitory control. Adolescents with high inhibitory control are generally able to inhibit negative behavioral responses [26], but may not yet know how to behave in specific situations that involve cyber aggression. Because susceptibility to peer pressure and inhibitory control may moderate the intervention effects in contrasting ways, we pose the following research question:

RQ: How do individual differences in susceptibility to peer pressure and inhibitory control moderate the expected intervention effects?

## Method

### Design

The study used a 2 (Peer Group Norms: Yes, No) x 2 (Reduced Accountability: Yes, No) factorial between-subjects design with four conditions. We randomly assigned participants to one of the four conditions: (1) peer group norms, (2) reduced accountability, (3) combination, and (4) control. We designed the conditions as online e-learning modules. The e-modules presented information, questions and tasks in a sequential order. At the first measurement directly after the intervention (Time 1), we used a messaging app paradigm and self-report measures to test the direct effects of the intervention. We used the same self-report measures again four weeks later (Time 2) to test the effects at four weeks post-intervention. Unfortunately, the COVID-19 outbreak interrupted data collection. The outbreak led to a national

lockdown and closing of all secondary schools in The Netherlands, where we conducted this research. As a result, not all participants could partake at the measurement four weeks post-intervention. Therefore, we tested the direct effects with the complete sample, whereas we tested the effects at four weeks post-intervention only with a partial sample.

## Participants

**Sample at Time 1.** The initial sample included 441 participants. As preregistered, we excluded participants who did not complete all measures ($n = 14$), who indicated to question the credibility of the messaging app paradigm ($n = 6$), and who deviated more than 2 standard deviations (SDs) of the mean of the time (in seconds) spent completing this first measurement ($n = 44$). This resulted in a final sample of 377 participants (53.6% boys, 46.1% girls, 0.3% did not indicate) between 11 and 16 years old ($M = 12.99$, $SD = 0.84$). Most participants attended vocational secondary education (53.2%), followed by senior general secondary education (38.3%), and preparatory university secondary education (8.5%).

**Sample at Time 2.** Due to the COVID-19 outbreak, the government in The Netherlands closed all secondary schools from 16 March 2020 onwards. Due to schools closing, 53.1% of participants ($n = 200$) could not partake at the second measurement four weeks post-intervention. We further excluded participants who deviated more than 2 SDs of the mean time spent completing this second measurement ($n = 10$). This resulted in a final sample of 167 participants (51.5% boys, 47.9% girls, 0.6% did not indicate) between 12 and 15 years old ($M = 12.96$, $SD = 0.68$). Most participants attended vocational secondary education (58.1%), followed by senior general secondary education (40.1%), and preparatory university secondary education (1.8%).

## Procedure

The study received ethical approval of the Ethics Committee Social Sciences of Radboud University (ref. ECSW-2019-138) and we pre-registered it at AsPredicted. The preregistration and anonymized data of the study are available on the Open Science Framework (OSF, https://bit.ly/2SYR1V1). Additionally, data management was in line with the data management protocol of Radboud University. The study took place in February and March 2020 during school hours. We recruited schools through information letters and phone calls. We provided workshops on social media netiquette (i.e., appropriate ways to behave on the internet) for students in return for their participation. We obtained active, written informed consent from schools, legal guardians, and participants themselves before participation. Participants completed the study via *Qualtrics* (a software program for online experiments) in two in-class sessions on a laptop or desktop PC.

The first session (Time 1) lasted approximately 30 minutes and consisted of four parts. First, participants completed a questionnaire containing demographics and questions about their WhatsApp use. Second, participants completed the intervention. Third, participants completed a messaging app paradigm we previously developed to test conformity to cyber aggression [44]. Fourth, participants completed a questionnaire containing self-report measures for conformity to cyber aggression, moderator variables, additional variables (i.e., intervention evaluation measures), and covariates.

The second session (Time 2) was conducted four weeks post-intervention and lasted approximately 5 minutes. It consisted of a questionnaire that included the same self-report measures for conformity to cyber aggression used at Time 1. A visualization of the study procedure is displayed in Fig 1.

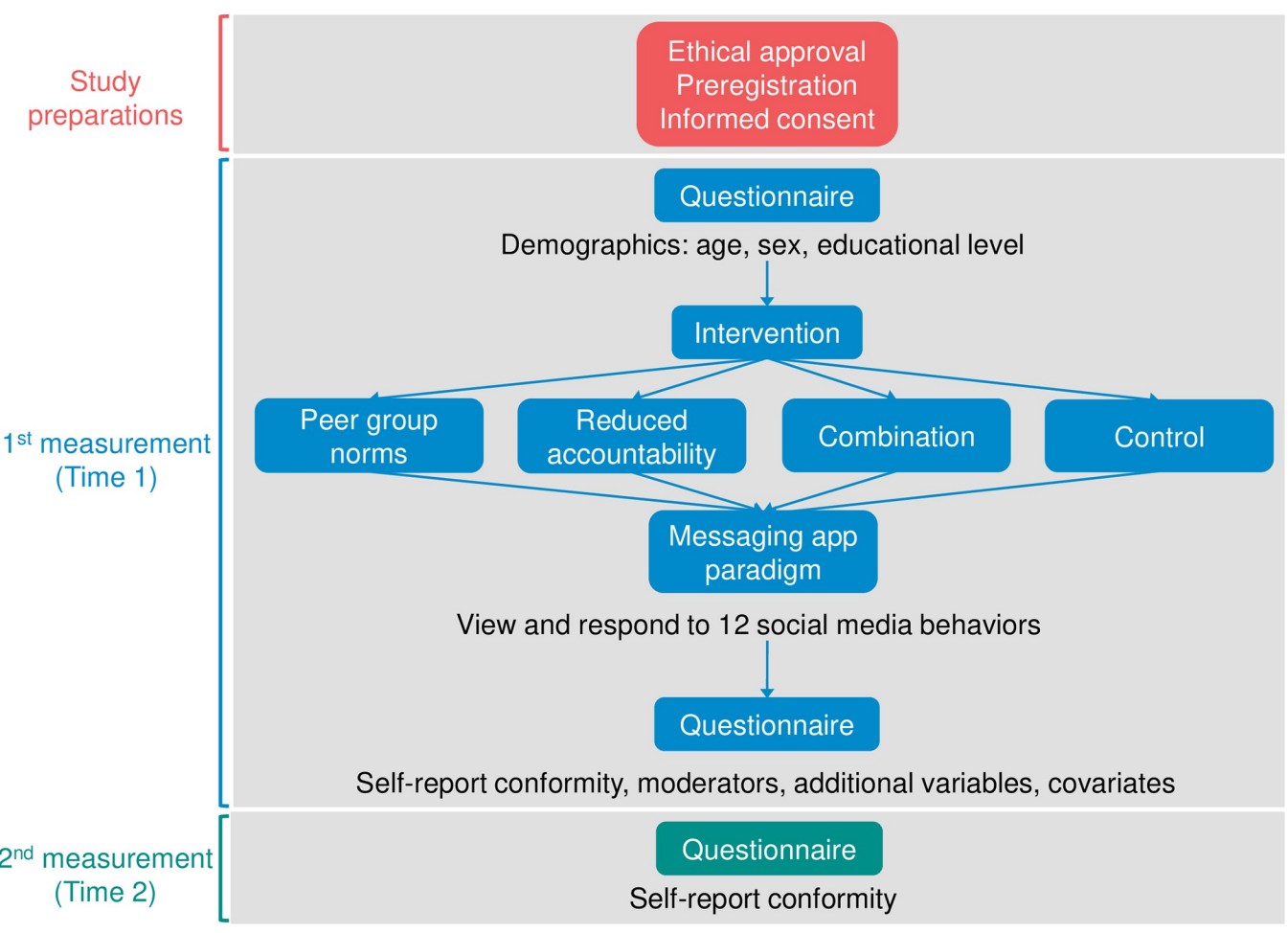

**Fig 1. Study procedure.**

## Conditions

**Peer group norms.**   To implement the information component in the peer group norms condition, we first informed participants about WhatsApp usage statistics. Adolescents must be able to recognize their behavior as cyber aggression to change it. Therefore, we then had participants read a description of the target behavior. This description included examples of different types of cyber aggression (e.g., impersonation, visual, written-verbal). Subsequently, participants were informed about the influence of peer group norms. Specifically, they learned how and why peer group norms increase conformity to cyber aggression. Finally, we informed participants about the general social norm on cyber aggression: Most people disapprove of cyber aggressive behaviors. We included this information to correct potential misperceptions about the behavior (in case adolescents thought that most peers considered cyber aggression appropriate).

To implement the motivation component, participants applied self-persuasion. Self-persuasion is a powerful technique to intrinsically motivate people, because with self-persuasion, people come up with personal arguments for why they would change their behavior. Such arguments are persuasive because they align with people's attitudes and beliefs [46, 54] and are driven by what an internal source (adolescents themselves) considers to be important as opposed to what external sources (peers, parents, or educators) find important [55].

Specifically, after being taught how peer group norms increase conformity, we asked participants to formulate two arguments for why it was important for them not to conform to cyber aggressive behaviors in messaging apps. To enhance participants' personal commitment [56], we displayed their self-generated arguments back to them on their screen, and we asked participants to confirm that these were the arguments that were important to them.

To implement the behavioral skills component, we instructed participants to formulate an implementation-intention for what they would do if their messaging app group engaged in cyber aggression [48–50]. Implementation-intentions are if-then plans in which people devise an action (the 'then') for when they find themselves in a particular situation (the 'if'). Implementation-intentions are powerful in helping people regulate their behavior [e.g., 49, 50, 57], because they stimulate people to replace automatic behavior with an alternative automatic response learned in the implementation-intention [48]. Specifically, participants completed the sentence: "If people in my messaging app group act in an unpleasant way towards people outside of the group, then . . .". To enhance personal commitment, participants' implementation-intention was displayed on their screen. Moreover, because repetition is central to the effectiveness of implementation-intentions [48], we asked participants to repeat their implementation-intention three times in their head, and type it in once more [57]. Table 1 depicts an overview of how we implemented the three IMB components in each of the intervention conditions.

**Reduced accountability.** Also in this condition, we first informed participants about WhatsApp usage statistics. Participants then read a description of the target behavior and learned about different types of cyber aggression. Subsequently, participants learned how reduced accountability increases conformity to cyber aggression. Finally, we informed participants about the general norm on cyber aggression, following the peer group norms condition. By informing participants about and increasing adolescents' awareness of reduced accountability, we aimed to counteract the potential influence of reduced accountability on conformity to cyber aggression.

To implement the motivation and behavioral skills components, participants received the same instructions to apply self-persuasion and formulate an implementation-intention as in the peer group norms condition. However, these instructions now followed the explanation of the reduced accountability determinant instead of the peer group norms determinant.

**Combination.** The combination condition combined the previously described intervention components. To implement the information, motivation and behavioral skills components of the IMB model, participants learned about the influence of peer group norms and reduced accountability, and formulated personal arguments and an implementation-intention for why they should not conform to cyber aggression on WhatsApp.

**Control.** In the control condition, participants completed questions and tasks about their social media use to ensure they would spend a similar amount of time on the study as participants in the other conditions. Specifically, we first informed participants about the definition of social media and different platforms (e.g., Instagram, YouTube, Netflix). Participants then answered questions about their social media use frequency, indicated their favorite platform and series on Netflix, and why they favored this platform or series. The control condition was comparable with the other conditions in terms of length, design, number of visuals, number of open/closed questions, and question phrasing.

## Measures

**Outcome variables.** We included several outcome variables to test the intervention effects. We used a messaging app paradigm [44] to assess conformity to cyber aggressive

**Table 1. Overview of the Implementation of the IMB components in each intervention condition.**

| Intervention component | Intervention conditions | | |
|---|---|---|---|
| | **Peer group norms** | **Reduced accountability** | **Combination** |
| **Information** | | | |
| General information | Basic information about WhatsApp: (1) it is the most popular platform among Dutch adolescents and (2) 96% of Dutch adolescents uses WhatsApp. | Basic information about WhatsApp: (1) it is the most popular platform among Dutch adolescents and (2) 96% of Dutch adolescents uses WhatsApp. | Basic information about WhatsApp: (1) it is the most popular platform among Dutch adolescents and (2) 96% of Dutch adolescents uses WhatsApp. |
| Context 1: target behavior | Description of the target behavior: on WhatsApp, people sometimes do nasty things **because their group members do so.** | Description of the target behavior: on WhatsApp, people sometimes do nasty things **because they feel free to do so.** | Description of the target behavior: on WhatsApp, people sometimes do nasty things **(1) because their group members do so, and (2) because they feel free to do so.** |
| Context 2: examples of target behavior | Examples of cyber aggression on WhatsApp: impersonation, visual, written-verbal, social exclusion. | Examples of cyber aggression on WhatsApp: impersonation, visual, written-verbal, social exclusion. | Examples of cyber aggression on WhatsApp: impersonation, visual, written-verbal, social exclusion. |
| Context 3: influence of determinant | Description of the determinant: **people value the opinion of their group members on WhatsApp, and, therefore, often do what the others in their group do. People conform to cyber aggressive behavior because they identify with their group members.** | Description of the determinant: **people feel anonymous on WhatsApp, and, therefore, feel more free to do nasty things. People conform to cyber aggressive behaviors because, on WhatsApp, they feel less responsible for their behaviors.** | Description of the determinant: **People conform to cyber aggressive behaviors for two reasons: (1) because they identify with their group members, and (2) because, on WhatsApp, they feel less responsible for their behaviors.** |
| Social norm | Description of the general social norm: hardly anyone approves of cyber aggressive behaviors on WhatsApp, but they occur nonetheless. | Description of the general social norm: hardly anyone approves of cyber aggressive behaviors on WhatsApp, but they occur nonetheless. | Description of the general social norm: hardly anyone approves of cyber aggressive behaviors on WhatsApp, but they occur nonetheless. |
| **Motivation** | | | |
| Self-persuasion | Self-generated arguments: If WhatsApp group members engage in cyber aggressive behaviors, why would you NOT join in? Please formulate 2 reasons for why you would not engage in cyber aggression if your group members do so. Type your reasons in the e-module. | Self-generated arguments: If WhatsApp group members engage in cyber aggressive behaviors, why would you NOT join in? Please formulate 2 reasons for why you would not engage in cyber aggression if your group members do so. Type your reasons in the e-module. | Self-generated arguments: If WhatsApp group members engage in cyber aggressive behaviors, why would you NOT join in? Please formulate 2 reasons for why you would not engage in cyber aggression if your group members do so. Type your reasons in the e-module. |
| Commitment enhancement | Participants' arguments were displayed back to them on their screen and participants had to confirm that these were their self-generated arguments. | Participants' arguments were displayed back to them on their screen and participants had to confirm that these were their self-generated arguments. | Participants' arguments were displayed back to them on their screen and participants had to confirm that these were their self-generated arguments. |
| **Behavioral skills** | | | |
| Implementation-intention | If-then plan: If WhatsApp group members engage in cyber aggressive behaviors, what would you do? Type your if-then plan in the e-module. | If-then plan: If WhatsApp group members engage in cyber aggressive behaviors, what would you do? Type your if-then plan in the e-module. | If-then plan: If WhatsApp group members engage in cyber aggressive behaviors, what would you do? Type your if-then plan in the e-module. |
| Commitment enhancement | Participants' if-then plans were displayed back to them on their screen, and participants had to confirm that this was their if-then plan. | Participants' if-then plans were displayed back to them on their screen, and participants had to confirm that this was their if-then plan. | Participants' if-then plans were displayed back to them on their screen, and participants had to confirm that this was their if-then plan. |
| Repetition 1 | Participants were asked to read and repeat their if-then plan 3 times in their head. | Participants were asked to read and repeat their if-then plan 3 times in their head. | Participants were asked to read and repeat their if-then plan 3 times in their head. |
| Repetition 2 | Participants were asked to type in their if-then plan once more. | Participants were asked to type in their if-then plan once more. | Participants were asked to type in their if-then plan once more. |

*Note.* Differences in the content between the conditions are underlined and in bold.

attitudes (i.e., how adolescents judged the appropriateness of cyber aggressive behaviors) and conformity to cyber aggressive intentions (i.e., whether adolescents intended to engage in cyber aggressive behaviors). We previously developed this messaging app paradigm and found

it suitable to test conformity to cyber aggressive attitudes [44]. However, we had not yet applied it to cyber aggressive intentions. Because we designed the current study with the intent to change behavior, we assessed both conformity to cyber aggressive attitudes and conformity to cyber aggressive intentions with this messaging app paradigm. Furthermore, we included self-report measures for conformity to cyber aggression.

We measured **conformity to cyber aggression in the messaging app paradigm** by using a scripted conversation (no real discussion) in an online environment visually similar to WhatsApp. We led participants to believe they would have a chat discussion with three peers from their class. We asked participants to indicate to what extent they found several cyber aggressive behaviors appropriate and to what extent they intended to engage in these behaviors. Before providing their answers, participants viewed their ostensible peers' responses. In reality, we manipulated the ostensible peers' responses to be positive towards cyber aggression (scores varying from 4 to 6 on a 6-point scale with 6 being most positive towards cyber aggression). The paradigm also included filler items on general media behaviors (e.g., playing a game on your phone). To increase the credibility of the conversation, half of the ostensible peers' responses on the filler items also included scores from 2 to 4.

We used items from the Cybervictimization Questionnaire [CYVIC, 58], assessing the extent to which youths experience aggression through mobile phones or the internet. Previous cyberbullying research has identified four main types of cyber aggression: impersonation (i.e., pretending to be someone else), visual (i.e., photos and videos), written-verbal (i.e., phone calls and comments), and social exclusion [58, 59]. We used 1 item for each cyber aggression behavior. For impersonation; "Impersonating someone else online, as if you are that person", for visual; "Take and disseminate an embarrassing photo of video of someone you know without his or her permission", for written-verbal; "Make fun of someone you know online, by making an offensive comment", and for social exclusion; "Not including a classmate in an online group or not letting him/her join a conversation".

We used these 4 items to assess conformity to cyber aggressive attitudes and conformity to cyber aggressive intentions. However, a pilot test for our study showed that using more than 4 target items (where the ostensible peers were positive towards cyber aggression) increased the chances that participants became suspicious about the true goal of the study. Therefore, we only let participants view one item about each type of cyber aggression (either the attitude or the intention item). We presented the target items to participants in a counterbalanced order so that all participants viewed 2 attitude and 2 intention items.

To measure conformity to cyber aggressive attitudes, participants indicated to what extent they thought each behavior was appropriate. Response categories ranged from 1 (*very inappropriate*) to 6 (*very appropriate*). To measure conformity to cyber aggressive intentions, participants indicated to what extent they intended to engage in the behavior. Response categories ranged from 1 (*definitely not*) to 6 (*definitely*). For both measures, we computed a difference score using the absolute difference between the participants' and the average scores of the ostensible classmates' responses. We reversed these scores so that higher scores indicated more conformity. The mean and standard deviation of these variables and those of the other key variables in this study are displayed in Table 2.

We also measured **self-reported conformity to cyber aggression** both directly after the intervention and four weeks post-intervention. We could only use the messaging app paradigm once, directly after the intervention. We could use the messaging app paradigm only once because we had to debrief participants after the first session to ensure that they knew that the peer responses in the WhatsApp conversation were not real.

We originally operationalized self-reported conformity to cyber aggression in two ways: (1) self-reported conformity to cyber aggressive attitudes and (2) self-reported conformity to

**Table 2. Means and standard deviations of key variables.**

| | Range | M | SD | N |
|---|---|---|---|---|
| **Outcome variables** | | | | |
| Experimental conformity to cyber aggressive attitudes | 0–4] | 1.68 | 1.14 | 353 |
| Experimental conformity to cyber aggressive intentions | [0–4] | 1.82 | 1.14 | 362 |
| Self-reported conformity to cyber aggression (Time 1) | [1–6] | 1.67 | 0.70 | 377 |
| Self-reported conformity to cyber aggression (Time 2) | [1–6] | 1.48 | 0.56 | 167 |
| **Moderator variables** | | | | |
| Susceptibility to peer pressure | [1–6] | 2.11 | 0.91 | 377 |
| Inhibitory control | [1–3] | 1.69 | 0.39 | 377 |
| **Additional variables** | | | | |
| Perceived importance not to conform | [1–6] | 4.14 | 1.60 | 377 |
| Perceived ease not to conform | [1–6] | 4.79 | 1.18 | 377 |
| Enjoyment of the intervention | [1–6] | 4.20 | 0.89 | 377 |

cyber aggressive intentions. To measure self-reported conformity to cyber aggression, we asked participants to what extent they thought that conforming to each of the four cyber aggression behaviors was appropriate (attitudes) and to what extent they intended to conform to each of the four cyber aggression behaviors (intentions). All response categories ranged from 1 to 6, with 6 being most positive towards (intending to engage in) cyber aggression. This operationalization resulted in two measures with 4 items each.

However, contrary to what we preregistered, we combined the two measures into one measure. We had three reasons for doing so. First, the items used to measure the two constructs originally stemmed from the same scale. We only adapted the wording to refer to either attitudes or intentions. Second, the correlation between the two constructs was high (.67 at Time 1 directly after the intervention and .62 at Time 2 four weeks post-intervention). The high correlation indicated that, on a methodological level, the items mostly measured the same underlying construct. Third, a factor analysis with all 8 items forced on one factor indicated that all items had acceptable factor loadings (all $\geq$ .595 at Time 1 directly after the intervention and $\geq$ .586 at Time 2 four weeks post-intervention). The items also had good reliability (Time 1: $\alpha$ = .85, Time 2: $\alpha$ = .82). Therefore, we combined the two constructs into one measure for self-reported conformity to cyber aggression.

**Moderator variables.** We measured **susceptibility to peer pressure** with Santor et al.'s [60] 6-item peer pressure scale. An example item was "My friends can push me into doing just about anything". Response categories ranged from 1 (*completely disagree*) to 6 (*completely agree*). The 6 items had acceptable reliability ($\alpha$ = .77).

We measured **inhibitory control** with the 12-item inhibition subscale of the Behavior Rating Inventory of Executive Function [BRIEF; 61]. The Dutch translation of the scale was used [62]. An example item was "I blurt things out". Response categories ranged from 1 (*never*) to 3 (*often*). The 12 items had good reliability ($\alpha$ = .84).

**Additional variables.** We included several additional variables to evaluate whether the self-persuasion and implementation-intentions tasks increased adolescents' motivation and behavioral skills. These additional variables served as intervention evaluation measures. We operationalized motivation as 'perceived personal importance not to conform to cyber aggression', because personal importance is considered a key aspect of motivation to perform behavior [63]. Specifically, we asked participants how important it was for them not to conform to cyber aggression (1 *[very unimportant]* to 6 *[very important]*). Furthermore, we operationalized behavioral skills as 'perceived ease not to conform to cyber aggression', because the belief

in one's capability to control behavior is often measured by asking people how easy or difficult it is to perform the target behavior [64]. Specifically, we asked participants how easy or difficult it would be for them not to conform to cyber aggression (1 *[very difficult]* to 6 *[very easy]*). Finally, we included a general measure for participants' enjoyment of the intervention. We wanted to know how participants evaluated the intervention because schools could potentially use it in the future. Specifically, we asked participants how (1) interesting, (2) important, (3) fun, (4) useless, and (5) difficult they found the intervention (1 *[completely disagree]* to 6 *[completely agree]*). The 5 items had acceptable reliability ($\alpha$ = .64). Items 4 and 5 were reverse coded before computing a mean score for enjoyment of the intervention. A higher score thus indicated a more positive evaluation of the intervention.

**Covariates.** We included sex, educational level, age, and school year to correct for possible confounding effects. Boys and lower-level students (vocational secondary education) are generally more likely to be involved in cyber aggression than girls and higher-level students [65]. Moreover, older adolescents are more likely to engage in cyber aggression than younger adolescents, likely because adolescents' online communication access increases as they age [39]. Furthermore, we asked participants how many days per week (*one* to *seven*) and hours per day (ranging from 1 [*less than half an hour per day*] to 9 [*more than 7 hours per day*]) they used WhatsApp, and how many WhatsApp groups they were a member of (ranging from 1 [*1 to 5 groups*] to 7 [*more than 30 groups*]). We included these as covariates, because adolescents' frequency of media use is a risk factor of cyber aggression [66, 67]. Lastly, we included a measure for the prevalence of cyber aggression in participants' social environment and a measure for how often participants had managed *not* to engage in cyber aggression. We controlled for these variables because previous exposure to cyber aggression predicts involvement in cyber aggression [39]. To do so, we asked participants how often in the past 30 days cyber aggression had occurred in their social environment, and how often they had managed not to conform to cyber aggression (1 *[never]* to 6 *[very frequently]*).

## Results

### Preparatory analyses

We analyzed the data in SPSS (Version 25.0). First, we conducted preparatory analyses. We generated descriptive statistics to describe participants' WhatsApp use and the general prevalence of cyber aggression in their social environment. Due to the hierarchical structure of the data (participants were nested within classes and schools), we then assessed the need to use a mixed-model approach to control for this nested data. We conducted a one-way independent ANOVA and chi-square tests to check whether randomization across conditions had been successful. We discuss the outcomes of these preparatory analyses below.

**Descriptive statistics.** We computed descriptive statistics for the measures included directly after the intervention (Time 1) and the measures included four weeks post-intervention (Time 2) due to the difference in sample size (see Table 3). Because the descriptive statistics were highly similar at both time points, we report the key statistics of the complete sample (Time 1) here in text. Most participants (99.7%) used WhatsApp. Of these, 98.9% belonged to at least one group on WhatsApp and most often these participants belonged to 1 to 5 active WhatsApp groups (50.8%). Of all participants, 89.1% used WhatsApp every day of the week, mostly one hour per day (34.8%), followed by 30 minutes per day (20.7%), less than 30 minutes per day (18.4%), and the remainder (26.1%) used WhatsApp 3 hours per day or more. Participants indicated that cyber aggression rarely occurred in their social environment ($M$ = 1.97, $SD$ = 0.92) and that, when it occurred, they resisted conforming quite often ($M$ = 4.18, $SD$ = 1.56).

**Table 3. Descriptive statistics directly after the intervention (Time 1, N = 377) and four weeks post-intervention (Time 2, N = 167).**

| | | Time 1 | Time 2 |
|---|---|---|---|
| Used WhatsApp | | | |
| | No | 0.3% | 0% |
| | Yes | 99.7% | 100% |
| Member of a group on WhatsApp | | | |
| | No | 1.1% | 0.6% |
| | Yes | 98.9% | 99.4% |
| Number of active WhatsApp groups | | | |
| | 0 | 0.3% | 0.6% |
| | 1 to 5 | 50.8% | 57.2% |
| | 6 to 10 | 33.2% | 30.2% |
| | 11 to 15 | 8.3% | 6.0% |
| | 16 to 20 | 4.0% | 4.2% |
| | 21 to 25 | 1.3% | 1.2% |
| | 26 to 30 | 0.5% | 0.6% |
| | 30 or more | 1.6% | 0% |
| WhatsApp use in days per week | | | |
| | 7 days | 89.1% | 88.6% |
| | 6 days | 3.5% | 4.2% |
| | 5 days or less | 7.4% | 7.2% |
| WhatsApp use in hours per day | | | |
| | Less than 30 minutes per day | 18.4% | 18.6% |
| | 30 minutes per day | 20.7% | 22.7% |
| | 1 hour per day | 34.8% | 31.1% |
| | 2 hours per day | 13.0% | 16.8% |
| | 3 hours per day | 8.2% | 6.0% |
| | 4 hours per day | 2.1% | 1.2% |
| | 5 or more hours per day | 2.8% | 3.6% |
| Prevalence in social environment | | 1.97 (0.92) | 1.91 (0.85) |
| Nonconformity to cyber aggression | | 4.18 (1.56) | 4.32 (1.62) |

**Clustering of the data.** To determine whether it was necessary to control for the nested data in our study, we first identified the most appropriate random effects structure. To do this, we compared the variance in the outcome variables for conformity to cyber aggression explained by (1) class and (2) school to an intercept-only model. We looked at the intraclass correlation (ICC) and a chi-square likelihood ratio test to decide whether it was necessary to include a random intercept for class and school. A random intercept for class and school would be added in a random effects model if the model fit improved significantly as indicated by a statistically significant chi-square likelihood ratio test.

Chi-square likelihood ratio tests showed that the variance in neither of the primary outcome measures could be attributed to differences between classes or schools. For all outcome variables (conformity to cyber aggressive attitudes, conformity to cyber aggressive intentions, and self-reported conformity to cyber aggression) the intercepts did not vary significantly across the different classes, all $\chi^2_{change's}$ (1) $\leq$ 3.14, all $p's \leq$ .076 (all ICC's $\leq$ .04), or schools, all $\chi^2_{change's}$ (1) $\leq$ 2.83, all $p's \leq$ .093 (all ICC's $\leq$ .02). Because there was no evidence of variation depending on either class or school, we proceeded to test the hypotheses with general linear modeling [68].

**Randomization checks.**   A one-way independent ANOVA and chi-square tests showed that the four conditions were comparable in terms of age, $F(3, 373) = 0.98$, $p = .403$, sex, $\chi^2(6) = 5.94$, $p = .430$, and educational level, $\chi^2(6) = 2.78$, $p = .836$. This indicated that randomization across conditions was successful.

## Analysis of covariance to test the hypotheses

To test the differences between the conditions in the outcome variables we conducted factorial ANCOVAs for each of the primary outcome variables (i.e., conformity to cyber aggressive attitudes, conformity to cyber aggressive intentions, and self-reported conformity to cyber aggression). We entered peer group norms (Yes, No) and reduced accountability (Yes, No) as the between-subjects factors in the ANCOVA model. Moreover, we created interaction terms for susceptibility to peer pressure and inhibitory control with the between-subject factors. We included all covariates discussed in the Methods section of this paper in the models.

The results did not show any differences between the conditions on the outcome variables directly after the intervention. There was no statistically significant effect of peer group norms on conformity to cyber aggressive attitudes, conformity to cyber aggressive intentions, or self-reported conformity to cyber aggression, all $F$'s $\leq 2.78$, all $p$'s $\geq .097$, all partial $\eta^2 \leq .01$ (see Table 4 for the mean of each outcome variable in the four conditions and Fig 2 for a visual representation of the differences between the conditions). Therefore, there was no support for H1 at Time 1 directly after the intervention.

There was also no statistically significant effect of reduced accountability on conformity to cyber aggressive attitudes, conformity to cyber aggressive intentions, or self-reported conformity to cyber aggression, all $F$'s $\leq 1.88$, all $p$'s $\geq .171$, all partial $\eta^2 \leq .007$. Thus, there was no support for H2 at Time 1.

Furthermore, there was no statistically significant interaction between peer group norms and reduced accountability on conformity to cyber aggressive attitudes, conformity to cyber aggressive intentions, or self-reported conformity to cyber aggression, all $F$'s $\leq 0.68$, all $p$'s $\geq .411$, all partial $\eta^2 \leq .003$. Thus, again, there was no support for H3 at Time 1.

Finally, susceptibility to peer pressure and inhibitory control did not moderate the relations for conformity to cyber aggressive attitudes, all $F$'s $\leq 1.95$, all $p$'s $\geq .164$, conformity to cyber aggressive intentions, all $F$'s $\leq 2.17$, all $p$'s $\geq .142$, or self-reported conformity to cyber aggression, all $F$'s $\leq 3.25$, all $p$'s $\geq .073$.

**Table 4.  Means and standard deviations for the outcome variables in each condition directly after the intervention (Time 1) and four weeks post-intervention (Time 2).**

| | Peer group norms | | |
|---|---|---|---|
| **Reduced accountability** | **Yes** | **No** | **Outcome variable** |
| | Time 1 | | |
| Yes | 1.83 (1.18) $n = 66$ | 1.65 (1.16) $n = 66$ | Conformity to attitudes in messaging paradigm |
| No | 1.80 (1.07) $n = 69$ | 1.81 (1.21) $n = 67$ | |
| Yes | 1.68 (1.15) $n = 71$ | 1.98 (1.08) $n = 68$ | Conformity to intentions in messaging paradigm |
| No | 1.96 (1.11) $n = 70$ | 1.85 (1.15) $n = 68$ | |
| Yes | 1.76 (0.79) $n = 73$ | 1.79 (0.70) $n = 71$ | Self-reported conformity |
| No | 1.83 (0.68) $n = 73$ | 1.82 (0.70) $n = 70$ | |
| | Time 2 | | |
| Yes | 1.43 (0.40) $n = 39$ | 1.49 (0.52) $n = 32$ | Self-reported conformity |
| No | 1.80 (0.85) $n = 32$ | 1.60 (0.51) $n = 24$ | |

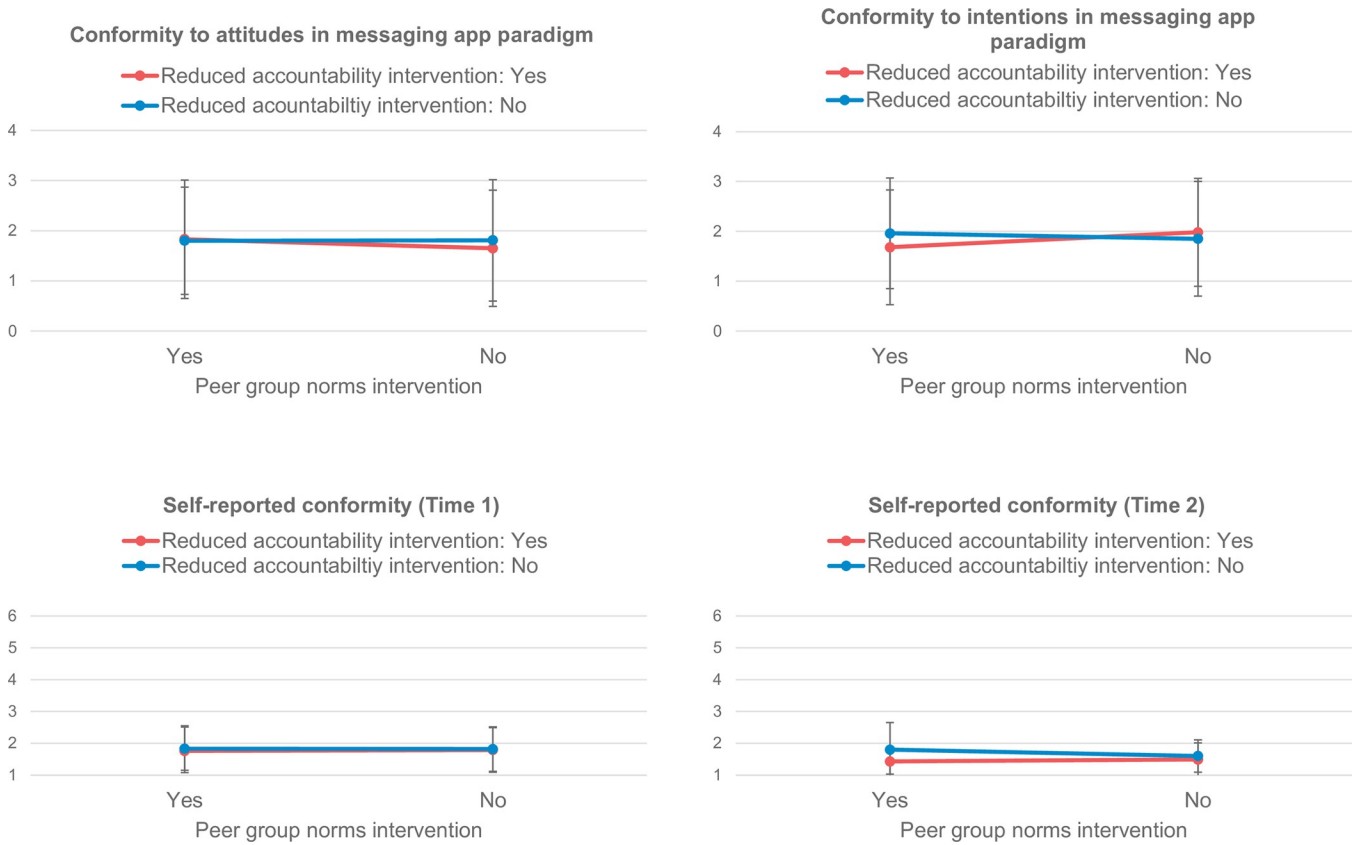

**Fig 2. Means for the outcome variables per condition (Error bars representing standard deviations).**

To test the hypotheses at four weeks post-intervention, we repeated the analysis for self-reported conformity to cyber aggression, but this time with self-reported conformity to cyber aggression as measured during the second session (Time 2). Again, there was no statistically significant effect of peer group norms on self-reported conformity to cyber aggression, $F(1, 106) = 0.32$, $p = .573$, partial $\eta^2 = .003$. There was also no statistically significant main effect of reduced accountability on self-reported conformity to cyber aggression, $F(1, 106) = 1.24$, $p = .268$, partial $\eta^2 = .012$. Furthermore, there was no statistically significant interaction between peer group norms and reduced accountability on self-reported conformity to cyber aggression, $F(1, 106) = 1.57$, $p = .213$, partial $\eta^2 = .015$. Finally, susceptibility to peer pressure and inhibitory control again did not moderate the relations, all $F's \leq 2.62$, all $p's \geq .109$. Therefore, we also did not find support for any of the hypotheses at four weeks post-intervention (Time 2).

## Analysis of variance to examine the evaluation of the intervention

The last set of analyses included one-way independent ANOVAs to examine differences between the conditions in how adolescents evaluated the intervention. Specifically, we tested differences between the conditions in perceived importance and ease not to conform to cyber aggression, and enjoyment of the intervention. The analyses' results showed no differences regarding importance, $F(3, 373) = 2.33$, $p = .075$. The peer group norms ($M = 3.99$, $SD = 1.60$), reduced accountability ($M = 3.85$, $SD = 1.64$), and combination condition ($M = 4.39$, $SD = 1.56$) did not increase perceived importance not to conform to cyber aggression compared to the control condition ($M = 4.27$, $SD = 1.53$). Moreover, we found no differences

regarding perceived ease not to conform to cyber aggression, $F(3, 373) = 1.81$, $p = .146$. The peer group norms ($M = 4.67$, $SD = 1.29$), reduced accountability ($M = 4.86$, $SD = 1.15$), and combination condition ($M = 4.98$, $SD = 1.05$) did not increase perceived ease not to conform compared to the control condition ($M = 4.65$, $SD = 1.21$). Based on the two measures for motivation and behavioral skills, we can, thus, conclude that the intervention techniques did not affect motivation and behavioral skills to not conform to cyber aggression respectively. Finally, the analyses showed no differences in enjoyment of the intervention between the four conditions, $F(3, 373) = 0.04$, $p = .989$. Participants evaluated the peer group norms ($M = 4.22$, $SD = 0.93$), reduced accountability ($M = 4.18$, $SD = 0.91$), combination ($M = 4.21$, $SD = 0.80$), and control condition ($M = 4.21$, $SD = 0.94$) equally positively.

## Discussion

This study reports a theory-based behavior change intervention specifically tailored to conformity to cyber aggression in early adolescents' messaging app groups. It is the first intervention effort grounded in recent theoretical insights on conformity to cyber aggression. Contrary to our expectations, we observed no differences between the intervention conditions on conformity to cyber aggression, neither directly after the intervention (all $p's \geq .097$) nor four weeks post-intervention (all $p's \geq .213$). Furthermore, susceptibility to peer pressure and inhibitory control did not moderate the expected relations (all $p's \geq .073$). Finally, the results suggest that the behavior change techniques used in the intervention did not affect adolescents' motivation ($p = .075$) or behavioral skills ($p = .146$). This study does not provide evidence that the behavior change intervention effectively reduces conformity to cyber aggression in early adolescents' messaging app groups. We discuss explanations for the lack of intervention effects below.

A first possible explanation is that it may be challenging to target theoretical mechanisms in applied interventions to change behavior. Previous research points to the theoretical relevance of peer group norms and reduced accountability in predicting conformity to cyber aggression [e.g., 44]. However, it is complicated to disrupt such existing mechanisms in practice. The intervention's underlying theoretical insights (e.g., peer group norms, reduced accountability) emerged from descriptive studies and experiments. For example, in our previous studies we examined the role of these processes in predicting conformity to cyber aggression. Perhaps it is not possible to manipulate these processes in an intervention. For example, regarding the influence of peer group norms, our findings may demonstrate that it is challenging to counteract existing peer relations in messaging app groups. Perhaps peer group norms play such an important role in adolescents' daily lives that their perceptions of these norms cannot be easily changed [18, 36]. Moreover, by teaching adolescents about the influence of reduced accountability on conformity, we had hoped to induce some sense of accountability of one's behavior in online contexts. However, teaching adolescents about the role of reduced accountability may not be sufficient to disrupt this process. Consequently, this may not have led to any changes in behavior.

A second potential explanation for why the intervention was ineffective is that the specific techniques we used did not work. The general lack of statistically significant differences between the conditions on the intervention evaluation measures supports this explanation. The intervention conditions did not increase participants' motivation or behavioral skills (operationalized as perceived importance and ease not to conform to cyber aggression) compared to the control condition. Important here is that participants reported relatively high levels of perceived importance and ease not to conform to cyber aggression overall. It thus seems like participants were already motivated and felt able to resist conformity. Perhaps our intervention could not further increase motivation and behavioral skills. Participants also reported

relatively low levels of conformity to cyber aggression and prevalence of cyber aggression in their social environment, which may imply that we targeted behavior that adolescents experience only rarely. It may then be difficult to further reduce such behavior. Nevertheless, even a single instance of cyber aggression (e.g., a private picture shared on social media) can have lasting harmful effects on victims [8]. That even single behaviors can have a lasting impact makes it important to remain committed to reducing conformity to cyber aggression.

A third possible explanation might be that the intervention evoked resistance. We stimulated adolescents to make autonomous decisions by applying self-persuasion and implementation intentions. Nevertheless, we explicitly instructed adolescents to perform the self-persuasion and implementation-intention tasks. Perhaps adolescents felt their behavior was driven by an external source (the researchers) rather than by themselves. Previous research has found that feeling obliged to perform a task increases resistance to behavior change [55]. Potentially, this could have cancelled out any intervention effects. However, we did not measure resistance. Therefore, this explanation is only speculative.

Based on our findings, we cannot definitively conclude that individual susceptibility to peer pressure and inhibitory control play no role in the effectiveness of interventions that reduce conformity to cyber aggression. Our results suggest that they do not. However, we cannot yet formulate a definite answer to this research question.

## Limitations, future research directions, and implications

Before discussing the implications of this study, we address four limitations and suggestions for improvement in future research. The first limitation concerns the potential mismatch between the implementation-intention task and our measure for conformity to cyber aggression. For the situation "If people in my messaging app group act in an unpleasant way towards people outside of the group. . ." some participants formulated an implementation-intention such as ". . . then I will mute the group", ". . . then I will exit the group", or ". . . then I will take a screenshot of the conversation and show this to my teacher". These are valuable implementation-intentions that help adolescents refrain from conforming to cyber aggression. However, participants could not perform these behaviors in the messaging app paradigm. For example, participants could not use any mute, exit, or screenshot functions. Technical strategies (such as muting the group) may be less complicated for adolescents to undertake in real-life. Such strategies could put adolescents less at risk of social punishment from peers compared to openly diverging from peer group norms [69]. Therefore, we believe that future studies should use an alternative measure for conformity to cyber aggression that better captures all possible responses to cyber aggression. For example, research could include these technical functions in a messaging app paradigm, giving adolescents the opportunity to actually mute a group chat or exit the group. Including these technical strategies may help capture the full breadth of potential responses to cyber aggression in adolescents' messaging apps.

The second limitation concerns the lack of sufficient repetition and opportunity for practice for adolescents' self-formulated implementation-intention. Through repetition, an implementation-intention can become habitual and lead to behavior change [48, 50, 70]. In our study, participants formulated and wrote down their implementation-intention, repeated it a few times for themselves, and wrote it down again. There was, thus, only little opportunity for repetition. Implementation-intentions also work well when people can practice them by applying them to relevant situations. For example, when students formulate an implementation-intention for how they will complete their homework, it works well to then ask them to do their homework [49]. However, our intervention did not offer adolescents the opportunity to practice their implementation-intention. Therefore, the implementation-intention task may have

been too brief to effectively change behavior. Future research could address this by conducting multiple intervention sessions in which participants practice their implementation-intention in several situations involving cyber aggression. This would ensure more repetition and opportunity for practice and could, potentially, improve intervention effectiveness.

The third limitation concerns the possibility of sampling bias. We required active consent from participants and their legal guardians for participation in the study. It is possible that (guardians of) participants who were previously involved in cyber aggression–either as a perpetrator or as a victim–were more reluctant to give consent than (guardians of) participants with little or no involvement in cyber aggression. The scores for prevalence of cyber aggression in adolescents' social environment were also relatively low, which means that the intervention possibly tried to reduce behavior that already occurred only to a small extent in this sample. Therefore, our intervention may not have targeted adolescents who could benefit most from an intervention to reduce conformity to cyber aggression in messaging apps. Future research should aim to draw representative samples from the population that include both adolescents with little or no involvement in cyber aggression and those who were previously a perpetrator or a victim. Research with such representative samples is important, because this might shed light on the effectiveness of interventions for different target populations. Moreover, it would allow for the development of more tailored interventions, where interventions fit participants' personal characteristics or previous experiences such as cyber aggression involvement [71]. Some research suggests that tailored interventions can be more successful in changing behavior compared to generic interventions [72]. Obtaining a more representative sample could be achieved by informing parents and participants more explicitly that they do not have to share any personal experiences and that their anonymity is guaranteed.

Finally, a fourth limitation is that our findings do not account for cultural influence. Recent research shows that cyber aggression is a worldwide phenomenon [73–75]. The mechanisms involved in cyber aggression (such as social identification and reduced accountability) are mostly universal [75, 76]. Yet, the prevalence of cyber aggression likely differs depending on specific learned cultural self-construals [73, 74, 76, 77]. In European cultures for example, people tend to have higher independent self-construal and view themselves as more separate from their social context. Contrarily, in many Asian cultures for example, people typically have higher interdependent self-construal and view themselves as more within a broader social context. Generally, higher independent self-construal is associated with higher involvement in cyber aggression perpetration [73, 74, 76, 77]. Moreover, higher interdependent self-construal is associated with higher perception of social responsibility and better inhibitory control [78].

Because adolescents with higher interdependent self-construal are expected to be already better at inhibiting their behavioral responses, we would expect the current intervention to be more effective for adolescents with a higher independent self-construal. Adolescents with a higher independent self-construal may benefit more from an intervention helping them to better regulate their behaviors. However, our sample only included participants from the same (European) country. Therefore, we cannot examine the role of cultural influence with our data. Our explanations of the potential effects of cultural influence are, thus, only speculative and remain of interest for future research. We recommend future research on conformity to cyber aggression in messaging apps to broaden the scope and compare the effectiveness of interventions cross-culturally. For example, data could be collected and compared in countries with interdependent self-construal cultures (e.g., Egypt, Pakistan, Japan) and independent self-construal cultures (e.g., the United States, Australia, Germany). Comparing these countries could shed light on the most effective intervention design for adolescents with different cultural backgrounds.

In conclusion, the findings of our study point to the complex relation between theory and practice: Although the theoretical underpinnings of desired behavior may be evident, manipulating these in an applied intervention can be complex. Our study was a first effort towards doing so. The study limitations and suggested research directions are valuable in informing future intervention efforts to reduce conformity to cyber aggressive behaviors in adolescents' messaging apps.

## Author Contributions

**Conceptualization:** Daniëlle N. M. Bleize, Doeschka J. Anschütz, Martin Tanis, Moniek Buijzen.

**Data curation:** Daniëlle N. M. Bleize.

**Formal analysis:** Daniëlle N. M. Bleize.

**Investigation:** Daniëlle N. M. Bleize.

**Methodology:** Daniëlle N. M. Bleize, Doeschka J. Anschütz, Martin Tanis, Moniek Buijzen.

**Project administration:** Daniëlle N. M. Bleize.

**Resources:** Daniëlle N. M. Bleize.

**Software:** Daniëlle N. M. Bleize.

**Supervision:** Doeschka J. Anschütz, Martin Tanis, Moniek Buijzen.

**Validation:** Daniëlle N. M. Bleize.

**Visualization:** Daniëlle N. M. Bleize.

**Writing – original draft:** Daniëlle N. M. Bleize.

**Writing – review & editing:** Daniëlle N. M. Bleize, Doeschka J. Anschütz, Martin Tanis, Moniek Buijzen.

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
