## [Decision Letter · Decision Letter 0]

23 Dec 2021

PONE-D-21-17000Developing and testing an online behavior change intervention to reduce conformity to cyber aggression in messaging appsPLOS ONE

Dear Dr. Bleize,

Thank you for submitting your manuscript to PLOS ONE. After careful consideration, we feel that it has merit but does not fully meet PLOS ONE’s publication criteria as it currently stands. Therefore, we invite you to submit a revised version of the manuscript that addresses the points raised during the review process.

Please see reviewer comments below and submit  your revised manuscript by 20th January 2022. If you will need more time than this to complete your revisions, please reply to this message or contact the journal office at plosone@plos.org. Please include the following items when submitting your revised manuscript:A rebuttal letter that responds to each point raised by the academic editor and reviewer(s). You should upload this letter as a separate file labeled 'Response to Reviewers'.A marked-up copy of your manuscript that highlights changes made to the original version. You should upload this as a separate file labeled 'Revised Manuscript with Track Changes'.An unmarked version of your revised paper without tracked changes. You should upload this as a separate file labeled 'Manuscript'.If applicable, we recommend that you deposit your laboratory protocols in protocols.io to enhance the reproducibility of your results. Protocols.io assigns your protocol its own identifier (DOI) so that it can be cited independently in the future. For instructions see: https://journals.plos.org/plosone/s/submission-guidelines#loc-laboratory-protocols. Additionally, PLOS ONE offers an option for publishing peer-reviewed Lab Protocol articles, which describe protocols hosted on protocols.io. Read more information on sharing protocols at https://plos.org/protocols?utm_medium=editorial-email&utm_source=authorletters&utm_campaign=protocols.

We look forward to receiving your revised manuscript.

Kind regards,

Muhammad Shahzad Aslam, Ph.D.,M.Phil., Pharm-D

Academic Editor

PLOS ONE

Journal Requirements:

Reviewers' comments:

Reviewer's Responses to Questions

**Comments to the Author**

1. Is the manuscript technically sound, and do the data support the conclusions?

Reviewer #1: Yes

Reviewer #2: Yes

2. Has the statistical analysis been performed appropriately and rigorously? 

Reviewer #1: Yes

Reviewer #2: Yes

3. Have the authors made all data underlying the findings in their manuscript fully available?

Reviewer #1: Yes

Reviewer #2: Yes

4. Is the manuscript presented in an intelligible fashion and written in standard English?

Reviewer #1: Yes

Reviewer #2: Yes

5. Review Comments to the Author

Reviewer #1: The manuscript is well written, contains detailed description of designed experiments, collected data, statistical evaluation and final explanations. I appreciate that the authors critically reviewed the results, including the fact that the initial hypotheses were not confirmed in the study. Such research is also necessary because it can raise new questions and lead to successive probably more detailed study.

The authors should add discussion about cultural influence. Would be the results in another country similar or different depending on cultural and social context?

Reviewer #2: The papers presents a very current topic, regarding users online behaviour in messaging apps, where authors proposed an intervention using the Information-Motivation-Behavioral Skills model to counteract cyber aggression.

It is unclear how was applied the Information-Motivation-Behavioral Skills model and how it is looks? Which is the mathematical model behind? Which are the variables of the model? The papers presents only the results of the IMB model, but the model description and implementation is lacking.

The Conclusion and implications section can be integrated together with study limitations and future research directions. The whole effort seems useless if in Conclusions the authors stated that "the intervention was not successful in reducing conformity to cyber aggression, our study provides important insights and recommendations for the development of future intervention efforts."

6. PLOS authors have the option to publish the peer review history of their article (what does this mean?). If published, this will include your full peer review and any attached files.

Reviewer #1: No

Reviewer #2: No

---

## [Author Response · Author response to Decision Letter 0]

24 Mar 2022

March 24, 2022

Muhammad Shahzad Aslam, Ph.D., M.Phil., Pharm-D

Academic Editor

PLOS ONE

Subject: Revision of manuscript PONE-D-21-17000

Dear dr. Muhammad Shahzad Aslam and reviewers,

Please find enclosed our revised manuscript PONE-D-21-17000R1 entitled “Developing and testing an online behavior change intervention to reduce conformity to cyber aggression in messaging apps”. We greatly appreciate the time and effort that was dedicated to reviewing our manuscript. We thank dr. Muhammad Shahzad Aslam and the reviewers for their thorough review and comments, which were very helpful in improving our manuscript.

On the next pages, we specify how we have addressed the reviewers’ comments and we describe the changes we have made. In our response, the reviewers’ comments are numbered, and our responses follow below in blue. The revised manuscript text is displayed in italics, with page numbers for reference. Changes to the references list are indicated at the end of this response letter.

Thank you again for your consideration of our revised manuscript. We look forward to your response.

Sincerely, on behalf of all authors

Daniëlle N.M. Bleize

 

Response to Reviewers’ Comments

We were pleased to read that the reviewers valued our study and found the manuscript well-written. The reviewers suggest several revisions to improve our manuscript, which we address on the next pages.

Reviewer #1

The manuscript is well written, contains detailed description of designed experiments, collected data, statistical evaluation and final explanations. I appreciate that the authors critically reviewed the results, including the fact that the initial hypotheses were not confirmed in the study. Such research is also necessary because it can raise new questions and lead to successive probably more detailed study.

1. The authors should add discussion about cultural influence. Would the results in another country be similar or different depending on cultural and social context?

Author response: We thank the reviewer for this suggestion. Based on recent literature on the role of cultural influence in adolescents’ cyber aggression, we believe that our intervention might be more effective for adolescents from cultures with higher independent self-construal compared to adolescents from cultures with higher interdependent self-construal. 

We have included two paragraphs in our discussion section that address the potential role of cultural influence and independent versus interdependent self-construals specifically. These paragraphs read as follows: 

“Finally, a fourth limitation is that our findings do not account for cultural influence. Recent research shows that cyber aggression is a worldwide phenomenon [57–59] and that the mechanisms involved in cyber aggression (such as social identification and reduced accountability) are mostly universal [59,60]. Yet, the prevalence of cyber aggression likely differs depending on specific learned cultural self-construals [57,58,60,61]. In European cultures for example, people typically have higher independent self-construal and view themselves as more separate from their social context. Contrarily, in many Asian cultures for example, people typically have higher interdependent self-construal and view themselves as more within a broader social context. Generally, higher independent self-construal is associated with higher involvement in cyber aggression perpetration [57,58,60,61]. Moreover, higher interdependent self-construal is associated with higher perception of social responsibility and better inhibitory control [62]. 

 Because adolescents with higher independent self-construal are expected to be involved in cyber aggression more often and adolescents with higher interdependent self-construal are expected to be already better at inhibiting their behavioral responses, we would expect that the current intervention is more effective for adolescents with a higher independent self-construal. These adolescents may simply benefit most from an intervention helping them to better regulate their behaviors. Nevertheless, given that the intervention was not effective, we would not currently expect the results to be any different for adolescents with higher interdependent self-construal. However, our sample only included participants from the same (European) country. Therefore, we cannot actually examine the role of cultural influence with our own data. Our explanations of the potential effects of cultural influence are, thus, only speculative and remain of interest for future research.” (p. 31-32, discussion)

Reviewer #2

The papers presents a very current topic, regarding users online behaviour in messaging apps, where authors proposed an intervention using the Information-Motivation-Behavioral Skills model to counteract cyber aggression.

1. It is unclear how was applied the Information-Motivation-Behavioral Skills model and how it looks? Which is the mathematical model behind? Which are the variables of the model? The papers presents only the results of the IMB model, but the model description and implementation is lacking.

Author response: We thank the reviewer for this remark. We used the Information-Motivation-Behavioral Skills model (IMB model) as a theoretical framework guiding the development of our intervention but did not explicitly test possible causal relationships between the variables of the IMB model. Therefore, we do not specify its mathematical model in our manuscript. However, we do agree with the reviewer that the description of the IMB model and how we implemented this in our intervention can be improved. To address the reviewer’s questions, we therefore made several changes or additions to the manuscript text. 

First, we explicated that the IMB model principally offers a theoretical approach to develop behavioral interventions and that we used this theoretical approach to guide the development of our own intervention (p. 6-7). Second, we improved the general description of the IMB model by explaining the three key components of the model (information, motivation, and behavioral skills) and how these are interrelated (p. 6-7). Third, we added a figure (Fig 1) of the IMB model as applied to the concepts in our study, with the hope that this better explains how the key components of the IMB model translate to the specific topic of conformity to cyber aggression in our intervention. This figure is included with our resubmission, is referenced in the manuscript text (p. 7), and is displayed below. Finally, we more explicitly described how each component of the IMB model was implemented in our intervention by rephrasing sections throughout the manuscript (p. 7, 11-16, 20).

Revisions were made throughout the sections ‘Systematically translating theoretical insights to an applied intervention’, and ‘Testing a behavior change intervention to reduce conformity in messaging apps’ in the introduction section, and throughout the method section (visible in the revised manuscript with changes and indicated with the references to page numbers above).

Fig 1. Information-Motivation-Behavioral Skills Model Applied to Conformity to Cyber Aggression.

2. The Conclusion and implications section can be integrated together with study limitations and future research directions. The whole effort seems useless if in Conclusions the authors stated that "the intervention was not successful in reducing conformity to cyber aggression, our study provides important insights and recommendations for the development of future intervention efforts."

Author response: As per suggestion of the reviewer, we have integrated the two sections ‘Limitations and future research directions’ and ‘Conclusion and implications’ into one section named ‘Limitations, future research directions, and implications’. We rephrased this section of the discussion, in particular the final paragraph stating the implications of our study. We feel that this paragraph now better reflects how our study contributes to research and practice considering its limitations and suggested directions for future research. The final paragraph now reads as follows: 

“In conclusion, the findings of this study point to the complex relation between theory and practice: Although the theoretical underpinnings of desired behavior may be evident, manipulating these in an applied intervention can be challenging. Our study was a first effort towards doing so and its limitations and suggested research directions are valuable in informing future intervention efforts to reduce conformity to cyber aggressive behaviors in adolescents’ messaging apps.” (p. 33, discussion)

 

Changes to the Reference List

The following references were added to the reference list (no other changes were made):

57. Barlett CP, Gentile DA, Anderson CA, Suzuki K, Sakamoto A, Yamaoka A, et al. Cross-cultural differences in cyberbullying behavior: A short-term longitudinal study. J Cross Cult Psychol. 2014;45(2):300–13. 

58. Leung ANM, Wong N, Farver JM. Testing the effectiveness of an e-course to combat cyberbullying. Cyberpsychology, Behav Soc Netw. 2019;22(9):569–77. 

59. Kowalski R, Limber SP, McCord A. A developmental approach to cyberbullying: Prevalence and protective factors. Aggress Violent Behav. 2019;45(March–April):20–32. 

60. Barlett CP, Seyfert LW, Simmers MM, Hsueh Hua Chen V, Cavalcanti JG, Krahé B, et al. Cross-cultural similarities and differences in the theoretical predictors of cyberbullying perpetration: Results from a seven-country study. Aggress Behav. 2021;47(1):111–9. 

61. Wright MF, Kamble S V., Soudi SP. Indian adolescents’ cyber aggression involvement and cultural values: The moderation of peer attachment. Sch Psychol Int. 2015;36(4):410–27. 

62. Shapka JD, Law DM. Does one size fit all? Ethnic differences in parenting behaviors and motivations for adolescent engagement in cyberbullying. J Youth Adolesc. 2013;42(5):723–38.

---

## [Decision Letter · Decision Letter 1]

12 Apr 2022

PONE-D-21-17000R1Developing and testing an online behavior change intervention to reduce conformity to cyber aggression in messaging appsPLOS ONE

Dear,

Thank you for submitting your manuscript to PLOS ONE. After careful consideration, we feel that it has merit but does not fully meet PLOS ONE’s publication criteria as it currently stands. Therefore, we invite you to submit a revised version of the manuscript that addresses the points raised during the review process.

Please submit your revised manuscript by 27 May 2022. If you will need more time than this to complete your revisions, please reply to this message or contact the journal office at plosone@plos.org. Please include the following items when submitting your revised manuscript:A rebuttal letter that responds to each point raised by the academic editor and reviewer(s). You should upload this letter as a separate file labeled 'Response to Reviewers'.A marked-up copy of your manuscript that highlights changes made to the original version. You should upload this as a separate file labeled 'Revised Manuscript with Track Changes'.An unmarked version of your revised paper without tracked changes. You should upload this as a separate file labeled 'Manuscript'.

We look forward to receiving your revised manuscript.

Kind regards,

Muhammad Shahzad Aslam, Ph.D.,M.Phil., Pharm-D

Academic Editor

PLOS ONE

Additional Editor Comments (if provided):

Kindly write the abstract in a structured manner such as background, objective, methodology, results and conclusion. no need to give these headings but the abstract must be comprehensive.

Please provide the heading of the theoretical background and give complete literature in context to the theoretical background. Please provide all hypothesis related to Information-Motivation-Behavioral Skills Model Applied to Conformity to Cyber Aggression. (Fig 1)

The abbreviation such as T1 and T2 is meaningless. Please revisit the abbreviations. If you want to use these then please illustrate them in a picture.

Please provide an illustration of the study design incomplete manner.

Please write the analysis name instead to use the term additional analysis OR Main analyses.

Please write the manuscript in a simple way. The manuscript was prepared in a complicated manner and was difficult to read and understand the analysis. Please revisit your figures and provide clear information.

Reviewers' comments:

Reviewer's Responses to Questions

**Comments to the Author**

1. If the authors have adequately addressed your comments raised in a previous round of review and you feel that this manuscript is now acceptable for publication, you may indicate that here to bypass the “Comments to the Author” section, enter your conflict of interest statement in the “Confidential to Editor” section, and submit your "Accept" recommendation.

Reviewer #1: All comments have been addressed

2. Is the manuscript technically sound, and do the data support the conclusions?

Reviewer #1: Yes

3. Has the statistical analysis been performed appropriately and rigorously? 

Reviewer #1: Yes

4. Have the authors made all data underlying the findings in their manuscript fully available?

Reviewer #1: Yes

5. Is the manuscript presented in an intelligible fashion and written in standard English?

Reviewer #1: Yes

6. Review Comments to the Author

Reviewer #1: The authors responded to all reviewers´ comments. They clarified the addressed issues. The manuscript reads well.

7. PLOS authors have the option to publish the peer review history of their article (what does this mean?). If published, this will include your full peer review and any attached files.

Reviewer #1: No

---

## [Author Response · Author response to Decision Letter 1]

10 Jun 2022

June 10, 2022

Muhammad Shahzad Aslam, Ph.D., M.Phil., Pharm-D

Academic Editor

PLOS ONE

Subject: Revision of manuscript PONE-D-21-17000R1

Dear dr. Muhammad Shahzad,

Please find enclosed our revised manuscript PONE-D-21-17000R1 entitled “Developing and testing an online behavior change intervention to reduce conformity to cyber aggression in messaging apps”. 

We appreciate the time and effort that you invested as academic editor to review our manuscript. We thank you for your additional comments to our manuscript. We believe these have helped us greatly in improving our manuscript. Also, we would like to express our gratitude for granting us sufficient time to work on this revision.

On the next pages, we outline how we have addressed the comments and we describe the changes we have made. In our response, the comments are numbered and our responses follow below (in blue). The revised manuscript text is in italics, with page numbers for reference. Changes to the reference list are indicated at the end of this response letter.

Thank you again for your consideration of our revised manuscript. We look forward to your response.

Sincerely, also on behalf of my co-authors,

Daniëlle N.M. Bleize

 

Response to Editor Comments

1. Kindly write the abstract in a structured manner such as background, objective, methodology, results and conclusion. No need to give these headings but the abstract must be comprehensive.

Author response: We thank the editor for this comment. We agree that the abstract was not fully comprehensive, particularly with regards to the methodology of our study. We have critically revised the abstract and believe that it now provides a better structured overview of our study. Following the editor's comment, we did not include headings in the abstract. However, we rephrased sections of the abstract to more explicitly indicate its structure. For example, we now explicitly state the objective of our study ("…the objective of the current preregistered study was to…", p.2), outline the study procedure ("…participants were first informed about…", p. 2), and include the analysis we used to examine our main hypotheses ("Factorial ANCOVAs revealed that…", p. 2). Moreover, we extended the methodology section of the abstract. We believe that the abstract (p. 2 of the manuscript) now more clearly explains how the intervention was developed and how the IMB model was implemented.

2. Please provide the heading of the theoretical background and give complete literature in context to the theoretical background. Please provide all hypothesis related to Information-Motivation-Behavioral Skills Model Applied to Conformity to Cyber Aggression. (Fig 1)

Author response: Following the editor's suggestion, we added a heading for the theoretical background. We revised the introduction and theoretical background and revisited the literature. We conducted a literature search to search for (recent) relevant articles that we might have missed in the previous version of our manuscript. Based on this, we added recent articles as well as some key literature to the theoretical background. Moreover, we extended the explanation of recent empirical findings in the theoretical background (for example regarding our previous experiments, on p. 7 of the manuscript).

After careful consideration, we have decided to omit the figure of the IMB model (previously Fig 1) from our manuscript. Fig 1 displayed the IMB-model applied to conformity to cyber aggression. However, we believe that this figure may cause confusion among the reader regarding the objective and design of our study, which we would like to prevent. We designed this intervention based on the IMB principles (that adolescents should be informed, motivated and able to perform behavior). We compared different versions of the intervention and all versions targeted all IMB elements. It was not our goal to measure or actually test the paths of the IMB framework itself. That is, we did not aim to examine the (inter)relations between the IMB elements. Rather, we aimed to compare the presence of such elements in an intervention with the absence of them (control condition). Therefore, we did not have any hypotheses about these relations. Moreover, we also lack the necessary measures to actually test these (inter)relations. We only measured perceived importance and ease to conform to be able to explore how the intervention was evaluated. These served more or less as proxy's for motivation and behavioral skills, but we did not include proper measures for these variables and included no measure for the information element. To prevent confusion regarding our study goals, we therefore consider it best to omit Fig 1 from the manuscript. We revised the description of the IMB framework and its implementation (p. 8-10, 15-20, and 25) to better clarify that we used the IMB framework to guide the development of the intervention conditions, but not to test the paths in the IMB framework itself. 

3. The abbreviation such as T1 and T2 is meaningless. Please revisit the abbreviations. If you want to use these then please illustrate them in a picture.

Author response: We acknowledge that the abbreviations were not meaningful in the previous version of our manuscript, because we did not specify their meaning before using them. Following the editors' comment, we decided to remove these abbreviations and instead use the terms 'Time 1' and 'Time 2'. In our manuscript we specify that 'Time 1' and 'Time 2' refer to the measurement times directly after the intervention and four weeks post-intervention respectively. We have also adapted these abbreviations in Fig 1 and Fig 2.

4. Please provide an illustration of the study design incomplete manner.

Author response: We have revisited all figures submitted with our manuscript, including the figure that displays our study design. We have created a new illustration that displays the study design as well as the study procedure that participants followed. The illustration displays the steps we took in preparing the study, and the two measurement sessions (Time 1 and Time 2) with their respective steps (e.g., questionnaires, intervention, messaging app paradigm). This illustration is displayed in Fig 1 (referred to on p. 15 of the manuscript and enclosed separately with our submission).

 

5. Please write the analysis name instead to use the term additional analysis OR Main analyses.

Author response: Following the editor's suggestion, we changed the headings of the analyses paragraphs in the results section of the manuscript (p. 30 and p. 33). The headings now include the name of the analysis that we conducted. We also identified any mention of the terms 'additional analysis' and 'main analysis' and replaced these with the name of the actual analysis that was used.

6. Please write the manuscript in a simple way. The manuscript was prepared in a complicated manner and was difficult to read and understand the analysis. Please revisit your figures and provide clear information.

Author response: We thank the editor for this comment. We have thoroughly rewritten the entire manuscript (see document Revised Manuscript with Track Changes). Some important changes we made include: (1) rewriting sentences from passive voice to active voice, (2) simplifying sentences by converting overly long sentences to shorter sentences, and (3) correcting for grammar and punctuation. In addition, we made many other changes throughout the paper to help the reader's understanding of our manuscript. For example, we revised the results section of the manuscript to clarify our findings. We hope that the manuscript now reads more easily and that the analyses are more clear. Finally, we have revisited our figures. This led to a new illustration for the study design and study procedure (Fig 1) and an updated illustration for the charts that display the means for the outcome variables per condition (Fig 2).

 

Changes to the Reference List

The following references were added to the reference list:

Cialdini RB, Goldstein NJ. Social influence: Compliance and conformity. Annu Rev Psychol. 2004;55:591–621.

Gardella JH, Fisher BW, Teurbe-Tolon AR. A systematic review and meta-analysis of cyber- victimization and educational outcomes for adolescents. Rev Educ Res. 2017;87(2):283–308.

Giletta M, Choukas-Bradley S, Maes M, Linthicum KP, Card NA, Prinstein MJ. A meta-analysis of longitudinal peer influence effects in childhood and adolescence. Psychol Bull. 2021;147(7):719–47.

Giumetti GW, Kowalski RM. Cyberbullying via social media and well-being. Curr Opin Psychol [Internet]. 2022;45:101314.

Hinduja S, Patchin JW. 2019 Cyberbullying data [Internet]. 2019. Available from: https://cyberbullying.org/2019-cyberbullying-data

Laursen B, Veenstra R. Toward understanding the functions of peer influence: A summary and synthesis of recent empirical research. J Res Adolesc. 2021;31(4):889–907.

Lozano-Blasco R, Cortés-Pascual A, Latorre-Martínez MP. Being a cybervictim and a cyberbully – The duality of cyberbullying: A meta-analysis. Comput Human Behav. 2020;111:106444.

Nguyen HTL, Nakamura K, Seino K, Vo VT. Relationships among cyberbullying, parental attitudes, self- harm and suicidal behavior among adolescents: Results from a school-based survey in Vietnam. BMC Public Health. 2020;20(1):1–10.

Olthof T, Goossens FA. Bullying and the need to belong: Early adolescents’ bullying-related behavior and the acceptance they desire and receive from particular classmates. Soc Dev. 2008;17(1):24–46.

Rudnicki K, Vandebosch H, Voué P, Poels K. Systematic review of determinants and consequences of bystander interventions in online hate and cyberbullying among adults. Behav Inf Technol [Internet]. 2022;1–18. 

Skilbred-Fjeld S, Reme SE, Mossige S. Cyberbullying involvement and mental health problems among late adolescents. Cyberpsychology. 2020;14(1).

Steer OL, Betts LR, Baguley T, Binder JF. “I feel like everyone does it”- adolescents’ perceptions and awareness of the association between humour, banter, and cyberbullying. Comput Human Behav [Internet]. 2020;108:106297. 

Wegge D, Vandebosch H, Eggermont S, Pabian S. Popularity through online harm: The longitudinal associations between cyberbullying and sociometric status in early adolescence. J Early Adolesc. 2016;36(1):86–107.

---

## [Decision Letter · Decision Letter 2]

4 Jul 2022

PONE-D-21-17000R2Developing and testing an online behavior change intervention to reduce conformity to cyber aggression in messaging appsPLOS ONE

Dear,

Thank you for submitting your manuscript to PLOS ONE. After careful consideration, we feel that it has merit but does not fully meet PLOS ONE’s publication criteria as it currently stands. Therefore, we invite you to submit a revised version of the manuscript that addresses the points raised during the review process. Please submit your revised manuscript by August 18, 2022. If you will need more time than this to complete your revisions, please reply to this message or contact the journal office at plosone@plos.org. Please include the following items when submitting your revised manuscript:A rebuttal letter that responds to each point raised by the academic editor and reviewer(s). You should upload this letter as a separate file labeled 'Response to Reviewers'.A marked-up copy of your manuscript that highlights changes made to the original version. You should upload this as a separate file labeled 'Revised Manuscript with Track Changes'.An unmarked version of your revised paper without tracked changes. You should upload this as a separate file labeled 'Manuscript'.

We look forward to receiving your revised manuscript.

Kind regards,

Muhammad Shahzad Aslam, Ph.D.,M.Phil., Pharm-D

Academic Editor

PLOS ONE

Reviewers' comments:

Reviewer's Responses to Questions

**Comments to the Author**

1. If the authors have adequately addressed your comments raised in a previous round of review and you feel that this manuscript is now acceptable for publication, you may indicate that here to bypass the “Comments to the Author” section, enter your conflict of interest statement in the “Confidential to Editor” section, and submit your "Accept" recommendation.

Reviewer #3: (No Response)

2. Is the manuscript technically sound, and do the data support the conclusions?

Reviewer #3: Yes

3. Has the statistical analysis been performed appropriately and rigorously? 

Reviewer #3: Yes

4. Have the authors made all data underlying the findings in their manuscript fully available?

Reviewer #3: Yes

5. Is the manuscript presented in an intelligible fashion and written in standard English?

Reviewer #3: Yes

6. Review Comments to the Author

Reviewer #3: In the following, there is a list of questions that the authors should answer and concentrate on:

1) In the abstract, this phrase “Our findings point to the complex relation between theory and practice and warrant future research to further examine the effectiveness of intervention tools to reduce conformity to cyber aggression” will make me confused about the meaning. Or it can be revised as “. Our findings point to the complex relation between theory and practice, and these can warrant future research to further examine the effectiveness of intervention tools which could be used to reduce the conformity of cyber aggression and behavioral intervention tools”. Please revised this article again and make it more readable.

2) Please draw these two pictures again, because these two are so blurred. The font should be revised to bigger one.

3) For the Limitation and future research direction part, the limitations are described so readable and obvious, but the future work and feasibility need more details. Especially, for the third limitation, if you could not find the proper participants, is there any alternative plan, for example, collecting data from the third-world countries, or using some mental health scales to avoid aggression trend?

4) In the Conclusion, please attach the p level after the phrase, when you are summarizing the findings. This would become more readable.

5) Considering to change the title. “Developing and testing an online behavior change intervention to reduce conformity to cyber aggression in messaging apps” would make readers think that this article can give them a practical and useful way to address cyber aggression.

7. PLOS authors have the option to publish the peer review history of their article (what does this mean?). If published, this will include your full peer review and any attached files.

Reviewer #3: **Yes: **Cheng Kang

---

## [Author Response · Author response to Decision Letter 2]

13 Jul 2022

July 13, 2022

Muhammad Shahzad Aslam, Ph.D., M.Phil., Pharm-D

Academic Editor

PLOS ONE

Subject: Revision of manuscript PONE-D-21-17000R2

Dear dr. Muhammad Shahzad Aslam,

Thank you for giving us the opportunity to submit a revised draft of our manuscript PONE-D-21-17000R3 entitled “Testing a first online intervention to reduce conformity to cyber aggression in messaging apps” for publication in PLOS ONE. 

We want to express our gratitude for the time and effort that you and the reviewers dedicated to providing feedback on our manuscript. We thank you for the insightful comments and suggestions for improvement to our manuscript. We have incorporated all of the suggestions made by the reviewers. 

On the next pages, we outline how we have addressed the suggestions and describe the changes that we made. In our response, the comments are numbered and our responses follow below (in blue). The revised manuscript text is in italics, with page numbers for reference (referring to the submitted document 'Manuscript'). Changes to the reference list are indicated on the final page of this response letter.

Thank you again for your consideration of our revised manuscript. We look forward to your response.

Sincerely, also on behalf of my co-authors,

Daniëlle N.M. Bleize

 

Reviewers' Comments to the Authors:

Reviewer 3

1. In the abstract, this phrase “Our findings point to the complex relation between theory and practice and warrant future research to further examine the effectiveness of intervention tools to reduce conformity to cyber aggression” will make me confused about the meaning. Or it can be revised as “. Our findings point to the complex relation between theory and practice, and these can warrant future research to further examine the effectiveness of intervention tools which could be used to reduce the conformity of cyber aggression and behavioral intervention tools”. Please revised this article again and make it more readable.

Author response: As suggested by the reviewer, we have revised this phrase in the abstract. The phrase now reads: 

"The findings from this first intervention effort point to the complex relationship between theory and practice. Our findings warrant future research to develop potential intervention tools that could effectively reduce conformity to cyber aggression." (p. 2)

Moreover, we agree with the reviewer's assessment that we can further improve the readability of our manuscript. Accordingly, throughout the manuscript, we made revisions to the text to make it more readable. The revisions include shortening and improving brevity of the manuscript text, eliminating passive voice, clarifications and many other things. We do not include all revisions in this response letter (because they are too many), but the revisions are visible in the submitted document titled 'Revised Manuscript with Track Changes'.

2. Please draw these two pictures again, because these two are so blurred. The font should be revised to bigger one.

Author response: Thank you for pointing this out. As suggested by the reviewer, we have redrawn both figures with a bigger font and improved image quality. 

3. For the Limitation and future research direction part, the limitations are described so readable and obvious, but the future work and feasibility need more details. Especially, for the third limitation, if you could not find the proper participants, is there any alternative plan, for example, collecting data from the third-world countries, or using some mental health scales to avoid aggression trend?

Author response: Thank you for pointing this out. The reviewer is correct that we did not adequately describe the future research directions for the third and fourth limitation. As suggested by the reviewer, we have added the following to the manuscript text to describe the future research directions related to the third limitation (that concerns the possibility of sampling bias):

"Future research should aim to draw representative samples from the population that include both adolescents with little or no involvement in cyber aggression and those who were previously a perpetrator or a victim. Research with such representative samples is important, because this might shed light on the effectiveness of interventions for different target populations. Moreover, it would allow for the development of more tailored interventions, where interventions fit participants' personal characteristics or previous experiences such as cyber aggression involvement [71]. Some research suggests that tailored interventions can be more successful in changing behavior compared to generic interventions [72]. Obtaining a more representative sample could be achieved by informing parents and participants more explicitly that they do not have to share any personal experiences and that their anonymity is guaranteed." (p. 31)

Furthermore, we have added the following to the manuscript text to describe the future research directions related to the fourth limitation (that concerns the lack of cross-cultural comparisons):

"We recommend future research on conformity to cyber aggression in messaging apps to broaden the scope and compare the effectiveness of interventions cross-culturally. For example, data could be collected and compared in countries with interdependent self-construal cultures (e.g., Egypt, Pakistan, Japan) and independent self-construal cultures (e.g., the United States, Australia, Germany). Comparing these countries could shed light on the most effective intervention design for adolescents with different cultural backgrounds." (p. 32)

4. In the Conclusion, please attach the p level after the phrase, when you are summarizing the findings. This would become more readable.

Author response: As suggested by the reviewer, we have added the p-levels after the relevant phrases in the conclusion (p. 28).

5. Considering to change the title. “Developing and testing an online behavior change intervention to reduce conformity to cyber aggression in messaging apps” would make readers think that this article can give them a practical and useful way to address cyber aggression.

Author response: We think this is an excellent suggestion. We have changed the manuscript title to: "Testing a first online intervention to reduce conformity to cyber aggression in messaging apps". We believe that this title better reflects that we test the effectiveness of a first intervention effort, without implying that this effort was effective and immediately implementable in practice. 

 

Changes to the Reference List

The following references were added to the reference list:

Jacobs NC, Völlink T, Dehue F, Lechner L. Online Pestkoppenstoppen: Systematic and theory-based development of a web-based tailored intervention for adolescent cyberbully victims to combat and prevent cyberbullying. BMC Public Health. 2014;14(396):1–16. 

Krebs P, Prochaska JO, Rossi JS. A meta-analysis of computer-tailored interventions for health behavior change. Prev Med (Baltim). 2010;51(3–4):214–21.

---

## [Decision Letter · Decision Letter 3]

25 Jul 2022

Testing a first online intervention to reduce conformity to cyber aggression in messaging apps

PONE-D-21-17000R3

Dear,

We’re pleased to inform you that your manuscript has been judged scientifically suitable for publication and will be formally accepted for publication once it meets all outstanding technical requirements.

Kind regards,

Muhammad Shahzad Aslam, Ph.D.,M.Phil., Pharm-D

Academic Editor

PLOS ONE

Additional Editor Comments (optional):

Reviewers' comments:

Reviewer's Responses to Questions

**Comments to the Author**

1. If the authors have adequately addressed your comments raised in a previous round of review and you feel that this manuscript is now acceptable for publication, you may indicate that here to bypass the “Comments to the Author” section, enter your conflict of interest statement in the “Confidential to Editor” section, and submit your "Accept" recommendation.

Reviewer #3: All comments have been addressed

2. Is the manuscript technically sound, and do the data support the conclusions?

Reviewer #3: Yes

3. Has the statistical analysis been performed appropriately and rigorously? 

Reviewer #3: Yes

4. Have the authors made all data underlying the findings in their manuscript fully available?

Reviewer #3: Yes

5. Is the manuscript presented in an intelligible fashion and written in standard English?

Reviewer #3: Yes

6. Review Comments to the Author

Reviewer #3: I have no more comment for this article and this article now looks more proper to be published on PLOS ONE.

7. PLOS authors have the option to publish the peer review history of their article (what does this mean?). If published, this will include your full peer review and any attached files.

Reviewer #3: No

---

## [Editor Report · Acceptance letter]

1 Aug 2022

PONE-D-21-17000R3 

Testing a first online intervention to reduce conformity to cyber aggression in messaging apps 

Dear Dr. Bleize:

I'm pleased to inform you that your manuscript has been deemed suitable for publication in PLOS ONE. Congratulations! Your manuscript is now with our production department. 

Kind regards, 

on behalf of

Dr. Muhammad Shahzad Aslam 

Academic Editor

PLOS ONE